# Fever and Pain in Italian Children: What Pediatricians Really Do

**DOI:** 10.3390/life15071048

**Published:** 2025-06-30

**Authors:** Giacomo Biasucci, Maria Elena Capra, Antonella Giudice, Delia Monopoli, Roberta Rotondo, Daniela Petracca, Cosimo Neglia, Beatrice Campana, Susanna Esposito

**Affiliations:** 1Pediatrics and Neonatology Unit, Guglielmo da Saliceto Hospital, 29121 Piacenza, Italy; g.biasucci@ausl.pc.it (G.B.); m.capra@ausl.pc.it (M.E.C.); 2Department of Medicine and Surgery, University of Parma, 43126 Parma, Italy; antonella.giudice@unipr.it (A.G.); delia.monopoli@unipr.it (D.M.); roberta.rotondo@unipr.it (R.R.); daniela.petracca@studenti.unipr.it (D.P.); negliamino@gmail.com (C.N.); beatricerita.campana@unipr.it (B.C.); 3Pediatric Clinic, Department of Medicine and Surgery, University of Parma, 43126 Parma, Italy

**Keywords:** analgesics, antipyretics, fever, pain, primary care pediatricians

## Abstract

**Background:** Fever and pain are among the most frequent symptoms in pediatric care, requiring timely and appropriate management. While evidence-based guidelines are available, adherence in real-world practice remains variable. This study aimed to explore the attitudes and prescribing behaviors of Italian Primary Care Pediatricians (PCPs) in the management of fever and pain, and to assess their alignment with current clinical recommendations. **Materials and Methods:** An anonymous, cross-sectional survey consisting of 30 multiple-choice questions was administered to 900 PCPs between 1 July and 30 October 2024. The questionnaire assessed therapeutic preferences, dosing strategies, and perceived knowledge gaps. Invitations were distributed via pediatric scientific societies and regional professional networks. **Results:** A total of 244 PCPs completed the survey (response rate 27.1%). The majority were aged over 55 years (72.1%), worked in urban settings (71.3%), and had more than 20 years of clinical experience (74.6%). Most respondents (77%) reported managing pediatric fever or pain on a daily basis. Paracetamol was the preferred first-line treatment for fever (95.9%), primarily due to its perceived safety (82.4%). Ibuprofen was favored by 51.6% of those who selected it for its greater effectiveness. The alternating use of paracetamol and ibuprofen for fever was never adopted by 49.6%, while 31.6% employed this strategy, believing it to be more effective. For pain, 67.6% used paracetamol and 26.2% used ibuprofen as first-line treatments; 15.2% reported alternating the two drugs. Correct dosage practices were followed by 63.9% for both medications, although 40.2% did not differentiate dosages between fever and pain management. **Conclusions:** While general trends showed alignment with current guidelines, notable inconsistencies were observed in drug selection, dosage, and the use of alternating therapies. These findings highlight a pressing need to improve the dissemination and implementation of pediatric fever and pain management guidelines among PCPs in order to reduce unsafe practices, avoid therapeutic errors, and prevent unnecessary strain on emergency care services.

## 1. Introduction

Fever and pain are among the most common symptoms in pediatric patients, often reflecting underlying infectious or non-infectious conditions that require accurate evaluation and appropriate therapeutic intervention [1,2]. Their effective management is essential not only for reducing discomfort and distress but also for preventing potential complications and improving clinical outcomes.

Fever is typically defined as a transient elevation in body temperature, commonly resulting from infections or inflammatory stimuli, and is generally considered present when the body temperature exceeds 38 °C (100.4 °F). It represents a physiological response that activates the immune system. The American Academy of Pediatrics (AAP) emphasizes that fever is a frequent reason for pediatric consultations and may originate from a broad range of clinical conditions [3]. In line with national recommendations, the Italian Pediatric Society has developed evidence-based guidelines to support clinicians in fever assessment and treatment. These include recommendations on temperature measurement methods based on age and setting, the use of paracetamol and ibuprofen as the only antipyretics recommended for pediatric use, and weight- and age-based dosing regimens. The key elements of these guidelines are summarized in Table 1 [4].

Pain, defined by the International Association for the Study of Pain as “an unpleasant sensory and emotional experience associated with actual or potential tissue damage, or described in terms of such damage,” is a complex, multidimensional experience influenced by sensory, cognitive, affective, and behavioral factors [5]. In children, pain assessment poses additional challenges due to varying levels of communication depending on the developmental stage and clinical condition. To address these challenges, several validated tools have been developed to quantify pediatric pain, considering subjective, behavioral, and physiological indicators [6]. The appropriate classification of pain—whether acute, chronic, or procedural—and its intensity is crucial for selecting effective therapeutic strategies, which often combine pharmacological and non-pharmacological approaches.

Acute pain is particularly frequent in pediatric practice, both in outpatient and hospital settings, most often associated with infections or minor trauma [7,8,9,10]. Concerns are growing regarding the possible long-term effects of unrelieved pain on the developing brain [11]. Despite increasing awareness and the availability of standardized treatment protocols, studies indicate that pediatric pain remains frequently under-recognized and undertreated [12]. The pharmacological treatment of acute pain in children generally involves two categories of medications: non-opioid analgesics, used for mild to moderate pain, and opioid analgesics, reserved for moderate to severe pain conditions, such as post-surgical pain or musculoskeletal trauma. The World Health Organization (WHO) outlines a stepwise approach to pediatric pain management according to severity, as shown in Table 2 [13].

In the outpatient setting, paracetamol and ibuprofen remain the only recommended medications for treating fever and mild to moderate pain in children. These agents are considered safe and effective when administered at recommended doses [14]. Paracetamol can be used from birth, while ibuprofen is indicated from three months of age. However, ibuprofen is contraindicated in specific clinical situations, such as dehydration, Kawasaki disease, and varicella. The oral route is preferred for its predictability in absorption, while the rectal route should be reserved for cases of vomiting. The alternating or combined use of paracetamol and ibuprofen is discouraged due to the potential for dosing errors and an increased risk of adverse effects, including acute kidney injury. The recommended dosages of these medications for the treatment of fever and pain are summarized in Table 3 [7,12,14].

Despite the availability of national and international guidelines, discrepancies persist in clinical practice regarding the choice of antipyretic or analgesic agent, dosage, and treatment strategy, particularly in the outpatient setting. In Italy, the National Health System provides medical assistance to all the population regardless of their medical insurance, so different drug prescriptions are not connected to insurance needs and do not lead to any economic burden for patients. Both paracetamol and ibuprofen are classified as “C class drugs”; that is, they can be bought without a medical prescription and their cost has to be paid by each citizen. Paracetamol and ibuprofen costs are comparable. This study aims to explore the current attitudes and prescribing behaviors of Italian Primary Care Pediatricians (PCPs) in the management of fever and pain, assess their adherence to evidence-based recommendations, and identify areas where further education or guideline dissemination may be needed.

## 2. Material and Methods

A cross-sectional survey was conducted using a digital questionnaire composed of 30 multiple-choice questions, designed to investigate the attitudes and clinical practices of Italian PCPs regarding the management of fever and pain in pediatric patients (Appendix A). The questionnaire aimed to assess clinical decision-making based on personal experience, adherence to current guidelines, and perceived knowledge gaps.

The survey was distributed to a total of 900 PCPs between 1 July and 30 October 2024. Participants were contacted through regional professional pediatric societies or via direct personal invitations. They had to be PCPs in current activity on the Italian territory. An initial invitation email contained a direct link to the online questionnaire, hosted on Google Forms, which ensured anonymous and secure data collection. All questions were mandatory, and the form could not be submitted unless fully completed. A reminder email was sent at the end of August 2024 to encourage participation.

At the time of the survey, Italy had approximately 6962 practicing PCPs, according to the Italian National Institute of Statistics (ISTAT) [15], meaning that the invited cohort represented more than 14% of the national total [15]. Of the 900 PCPs invited, 244 completed the questionnaire, yielding a participation rate of 27.1%. The sample was representative in terms of age and gender distribution [16]. Measures were in place to ensure that each respondent could only complete the survey once.

The questionnaire explored several key domains: the frequency of managing pediatric fever and pain; first-line pharmacological choices; preferred formulations and dosing regimens; knowledge of the pharmacodynamics of paracetamol and ibuprofen; confidence in their efficacy and safety; observed adverse effects; and perceived educational needs. Questions were written in clear and accessible language to maximize response accuracy and completion rate.

Based on the sample size, the survey had a margin of error of ±2.55%, allowing for statistically representative insights with a 95% confidence level. Data analysis was performed using STATA^®^ software (version 12, College Station, TX, USA). All responses were reported as absolute frequencies and percentages. For multiple-choice questions, percentages were calculated using the total number of respondents as the denominator. Group comparisons were conducted using the chi-square test (for expected frequencies ≥5) or Fisher’s exact test (for expected frequencies <5). A *p*-value of less than 0.05 was considered statistically significant.

## 3. Results

### 3.1. Population Characteristics

A total of 244 Primary Care Pediatricians (PCPs) completed the survey within the designated timeframe. Respondents were distributed across age groups as follows: 3.7% (*n* = 9) were aged 25–34 years, 10.3% (*n* = 25) were 35–44 years, 13.9% (*n* = 34) were 45–54 years, 38.5% (*n* = 94) were 55–64 years, and 33.6% (*n* = 82) were over 65 years of age. In terms of professional experience, the majority (74.6%) reported more than 20 years in clinical practice, followed by 13.5% with 11–20 years, 5.7% with 5–10 years, and 6.2% with less than 5 years.

Geographically, responses were received from PCPs working in 17 different Italian regions, with the highest concentrations reported in Emilia-Romagna (24.2%) and Puglia (22.5%). In terms of regional distribution, 59.3% of respondents practiced in Northern or Central Italy, while 40.7% worked in Southern regions or the islands. The majority of participants (71.3%) reported practicing in urban areas, 21.3% in suburban areas, and 4% in rural settings.

### 3.2. Paracetamol and Ibuprofen Prescription for Fever Treatment

Regarding the frequency of patient encounters for fever or pain, the majority of respondents (77%, *n* = 188) reported managing such cases on a daily basis. A minority (2.5%, *n* = 6) indicated seeing these cases monthly.

Paracetamol was the overwhelmingly preferred first-line treatment for fever, with 95.9% of PCPs (*n* = 234) indicating its routine use. A small number of respondents (1.6%, *n* = 4) reported using an alternating regimen of paracetamol and ibuprofen, while only one respondent (0.4%) reported using ibuprofen as their first-line option.

Among those who preferred paracetamol, 82.4% (*n* = 201) cited safety as the primary reason for their choice, 13.9% (*n* = 34) indicated perceived greater efficacy, and 1.6% (*n* = 4) referred to its lower cost.

Conversely, 51.6% of PCPs (*n* = 126) considered ibuprofen more effective and thus preferred it as a first-line treatment for fever, while 44.3% (*n* = 108) explicitly stated that they did not prefer ibuprofen for this indication. The reasons behind the choice of ibuprofen as a first-line agent, stratified by years of clinical experience, are illustrated in Figure 1.

### 3.3. Paracetamol and Ibuprofen Prescription for Pain Treatment

When asked about their preferred first-line treatment for managing pain in pediatric patients, 67.6% of PCPs (*n* = 165) reported using paracetamol, while 26.2% (*n* = 64) indicated a preference for ibuprofen. Notably, all PCPs who selected ibuprofen as their first-line option had more than 10 years of clinical experience (Figure 2).

Regarding the rationale for prescribing paracetamol, the majority of respondents (75%) stated that they considered it a safer option for children. A smaller proportion (11.1%) reported choosing paracetamol because they believed it to be more effective. Conversely, 8.2% indicated that they did not use paracetamol as a first-line treatment for pain; this subgroup included 12.1% of pediatricians with 11–20 years of experience and 8.8% with over 20 years of experience.

In contrast, ibuprofen was prescribed as the first-line treatment for pain primarily due to its perceived higher efficacy, as reported by 70.9% of those who preferred it. A smaller fraction (4.9%) indicated that their choice depended on the specific type of pain being treated, while only three respondents cited safety as a deciding factor. Just one participant considered ibuprofen and paracetamol to be equivalent in efficacy.

Regarding the use of combination or alternating therapy with paracetamol and ibuprofen for pain management, 52.9% of PCPs (*n* = 129) reported that they never adopted this strategy. Among those who did, 31.6% cited perceived superior efficacy as the reason, 2.9% considered it a safer approach, 0.8% used it at the request of parents, and another 2.9% cited other reasons.

### 3.4. Dosage and Formulation Prescribed

When asked about formulation preferences for ibuprofen prescriptions, 56.1% of PCPs (*n* = 137) reported favoring the pediatric formulation. This preference was most common among those with over 20 years of experience (58.2%), followed by those with 0–10 years (55.2%) and 11–20 years (45.5%). Conversely, 31.6% (*n* = 77) indicated having no preference regarding the formulation, while 8.2% (*n* = 20) did not prefer the pediatric preparation. A minority (3.3%, *n* = 8) opted for the lysine salt form of ibuprofen.

Regarding paracetamol dosing, 63.9% of respondents (*n* = 156) prescribed the recommended dose of 10–15 mg/kg every 4–6 h, with a similar distribution across all experience levels. A higher dosage (15–20 mg/kg) was prescribed by 34.4% (*n* = 84), and 1.6% (*n* = 4) provided unclear or non-standard responses. The complete dosing data by years of experience are presented in Table 4.

For ibuprofen, 63.9% (*n* = 156) of PCPs reported prescribing 5–10 mg/kg every 6–8 h, consistent with guideline recommendations, while 35.2% (*n* = 86) prescribed a higher dosage of 10–15 mg/kg. Again, 0.8% (*n* = 2) provided unclear answers. The full details are summarized in Table 5.

Notably, 42.6% of respondents (*n* = 104) indicated that they adjust the dosage of antipyretics and analgesics based on the severity of symptoms, with this practice being more frequent among PCPs with over 20 years of clinical experience (41.8%). When asked whether they differentiate between dosages for fever and pain, 40.2% (*n* = 98) reported that they do not—an approach that may reflect a lack of awareness regarding the need for higher initial doses of paracetamol in pain management compared to fever.

### 3.5. Perceived Knowledge, Management, and Side Effects of Paracetamol and Ibuprofen

When asked to self-assess their understanding of pharmacological mechanisms, 64.3% of respondents (*n* = 157) stated that they had a good knowledge of paracetamol’s mechanism of action, while 67.2% (*n* = 164) reported the same for ibuprofen.

In terms of confidence in the clinical efficacy of paracetamol for both fever and pain, 35.2% of PCPs (*n* = 86) declared they were very confident, 41% (*n* = 100) were confident, and 22.1% (*n* = 54) expressed confidence only in its use for fever, not for pain.

Confidence in ibuprofen’s efficacy was slightly higher: 25.4% (*n* = 62) of respondents were very confident and 58.6% (*n* = 143) reported being confident in its effectiveness for both fever and pain. A smaller percentage indicated partial confidence: 10.7% (*n* = 26) felt confident using it for pain but not for fever, 2.9% (*n* = 7) the opposite, and 2.5% (*n* = 6) stated that they were not confident in its effectiveness.

Adverse events associated with paracetamol were reported as rare by 56.6% (*n* = 138) and never encountered by 40.6% (*n* = 99). The most frequently observed side effects were allergic reactions (35.2%), followed by gastrointestinal disturbances (23%) and hepatic complications (13.9%). Only a small proportion (23.8%) reported no adverse effects at all (Table 6).

For ibuprofen, adverse effects were reported as rare by 62.7% (*n* = 153) and occasional by 27.9% (*n* = 68). Gastrointestinal symptoms were by far the most commonly observed (58.6%), followed by allergic reactions (32.4%). Other side effects, including hepatic and renal complications, were reported at lower frequencies. Only 4.1% of respondents stated that they had never observed any side effects linked to ibuprofen use.

Regarding educational needs, the majority of PCPs expressed a desire for further training. The most requested areas were dosing guidelines (25.4%) and safety considerations (24.2%), followed by the management of adverse effects (19.3%). Fewer respondents requested updates on mechanisms of action (12.7%) or efficacy (8.6%). A small number (6.1%) indicated no need for further training, while 2.9% expressed interest in all the proposed topics.

When asked about preferred methods for continuing education, 37.3% (*n* = 91) selected online courses or webinars, 36.9% (*n* = 90) preferred in-person workshops or seminars, and 23.4% (*n* = 57) opted for reading journals or scientific articles. Alternative methods, such as newsletters or database alerts (e.g., PubMed), were rarely chosen (<1%).

### 3.6. Analysis of Results According to Geographical Working Area of Respondents

Subgroup analysis based on geographical location revealed several notable differences in perceptions and prescribing behaviors among PCPs working in Northern/Central versus Southern Italian regions and islands.

Regarding the perceived efficacy of paracetamol, 15.6% of PCPs practicing in Northern and Central regions stated that they were confident in its use for treating fever but not for pain. This proportion rose to 31.1% among respondents from Southern regions and islands. Similarly, confidence in ibuprofen as a treatment for pain—but not for fever—was reported by 7.1% of PCPs in the north, compared to 15.5% in the south (*p* < 0.01), indicating a regional disparity in attitudes toward the drug’s indications.

Interest in further education also varied geographically. In Northern and Central Italy, 46.1% of participants expressed a desire for additional training on the use of analgesics and antipyretics in children, compared to 63.1% in Southern regions and islands, suggesting a higher perceived educational need in the latter group.

Differences were also observed in dosing practices, particularly in rural areas. Two-thirds of PCPs practicing in rural settings reported prescribing paracetamol at 15–20 mg/kg every 4–6 h—higher than guideline-recommended doses for fever—while one-third reported prescribing 10–15 mg/kg (*p* < 0.05 compared to suburban and urban areas). Similarly, 61.1% of rural practitioners prescribed ibuprofen at 10–15 mg/kg every 6–8 h, again exceeding standard recommendations. This contrasts with 38.5% in suburban and 31.6% in urban areas prescribing the same higher dose (*p* < 0.01), suggesting that location may influence clinical decision-making, potentially due to differing access to resources, training, or patient expectations.

## 4. Discussion

This survey revealed that the overwhelming majority of Italian PCPs manage fever and pain in children on a daily basis, with paracetamol being the preferred first-line treatment for both conditions. More than 95% of respondents reported prescribing paracetamol for fever, primarily due to its perceived safety, and 67.6% selected it as the first choice for pain management. However, ibuprofen was more frequently preferred for pain than for fever, particularly by more experienced physicians, who cited superior efficacy as the rationale. The alternating use of paracetamol and ibuprofen, though discouraged by national guidelines, was still practiced by approximately one-third of respondents for both fever and pain, often based on a belief in greater effectiveness [17,18,19].

The demographic structure of the sample (dominated by pediatricians over 55 years of age with more than two decades of clinical practice) reflects the aging workforce within Italy’s pediatric primary care sector. This group may rely heavily on longstanding habits or clinical experience, which could partly explain variations in adherence to guideline-recommended practices. The relatively limited representation of younger PCPs may also highlight a need for generational renewal in the profession.

Geographic and environmental differences also emerged. PCPs working in rural areas reported a higher-than-recommended dosing of both paracetamol and ibuprofen, a trend that may be linked to differences in continuing medical education opportunities or access to updated clinical guidance. This behavior, while possibly intended to ensure therapeutic effectiveness, poses safety risks and highlights the need for targeted training. Furthermore, 40.2% of PCPs reported using the same dosages for fever and pain, despite the evidence that higher initial doses of paracetamol may be needed to effectively manage pain, pointing to suboptimal differentiation in clinical practice.

Confidence in the safety and efficacy of both drugs was generally high. Paracetamol was more often associated with safety, while ibuprofen was more strongly associated with efficacy—particularly for pain with inflammatory components, such as pharyngotonsillitis. Reported adverse events were rare for both drugs. Nonetheless, gastrointestinal issues were commonly linked to ibuprofen use, consistent with the existing literature, and a small but concerning proportion of respondents prescribed ibuprofen at dosages exceeding guideline recommendations.

Educational needs were clearly expressed. Dosing and safety were the most frequently cited areas in which respondents desired further training, followed by the management of adverse effects. A notable geographical disparity was observed, with PCPs in Southern Italy and the islands expressing a greater perceived need for educational updates than their Northern and Central counterparts.

Despite the informative value of the results, this study has several limitations that should be acknowledged. First, the response rate of 27.1%, although comparable to other physician surveys [19,20,21], limits the generalizability of the findings and raises the possibility of response bias. It is plausible that pediatricians with a stronger interest in fever and pain management or those more engaged in continuing education were more likely to respond, potentially skewing the results toward more guideline-aware behaviors. Second, all data were self-reported and collected through an online questionnaire, which introduces inherent risks of recall bias and social desirability bias. Respondents may have overreported adherence to recommended practices or underreported the use of non-guideline-based approaches, such as alternating antipyretics. Additionally, the study design did not allow for verification of actual prescribing behavior through clinical records or pharmacy data. Another limitation lies in the limited exploration of contextual factors. Regional variation in responses suggests that socio-economic, cultural, and organizational influences likely play a role in shaping prescribing habits. However, the survey did not investigate these aspects in depth, nor did it assess the availability of local training resources or regional differences in access to pediatric care. The cross-sectional nature of the study also precludes any conclusions about causality or changes in practice over time. Moreover, the underrepresentation of younger PCPs, particularly those under 35 years of age, limits insight into the perspectives and training needs of the emerging pediatric workforce. Despite these limitations, the study also presents several important strengths. It is, to our knowledge, the first nationwide Italian survey specifically designed to assess real-world clinical practices and perceptions of fever and pain management among Primary Care Pediatricians. The survey instrument was comprehensive yet accessible, covering key domains, including therapeutic preferences, dosage practices, adverse effect monitoring, and perceived educational needs. The sample, while limited, was geographically diverse, with responses from 17 regions, and included urban, suburban, and rural contexts, thereby offering a valuable cross-sectional snapshot of clinical behaviors across the country.

The survey also highlighted discrepancies between guideline recommendations and actual practice, particularly in the use of alternating therapy and in dosing habits across different settings [22,23]. These insights are critical for informing future educational strategies, policy updates, and national consensus initiatives. By identifying specific knowledge gaps and regional trends, the study offers actionable evidence that can support the development of targeted interventions aimed at harmonizing pediatric fever and pain management practices in Italy. Overall, the study serves as a foundational step toward optimizing clinical approaches in outpatient pediatrics and underscores the need for continuous education, updated guidelines, and further research to support safe, effective, and evidence-based care.

We hope that the result of our survey may lay a basis for updating current fever and pain management in children guidelines. We plan to discuss the results of our survey with a panel of experts in the field so as to prepare a consensus statement on this topic. When combined, these results highlight the necessity of more organized, easily available continuing medical education programs, as well as an improved dissemination of clinical guidelines. Updated guidelines and a national consensus statement could support standardized prescription procedures and guarantee a more reliable, evidence-based treatment of pediatric outpatients’ fever and pain. Future investigations should examine the influence of the contextual and systemic elements on clinical decision-making, involve pediatricians working in hospitals, and validate these findings through larger-scale or longitudinal studies. In Italy, the Italian Pediatric Emergency and Urgency Society (Società Italiana di Medicina di Emergenza e Urgenza Pediatrica, SIMEUP) organizes seminars and workshops on pain management and emergency settings within the PIPER (Pain In Pediatric Emergency Room) project. The Italian Pediatrics Society (Società Italiana di Pediatria, SIP) has a working group on palliative care and pain management for terminally ill patients. However, structured and nation-wide projects targeted to PCPs to manage pediatric pain in outpatients’ setting are still to be implemented.

## 5. Conclusions

This study provides a detailed overview of the current clinical attitudes and prescribing practices of Italian PCPs regarding the management of fever and pain in children. The findings confirm the central role of these symptoms in everyday pediatric care and highlight a generally good adherence to national guidelines—particularly with respect to the first-line use of paracetamol for fever management. However, significant variability remains in the use of ibuprofen, in the preference for alternating regimens, and in the dosing strategies employed for both antipyretics and analgesics. Notably, prescribing behaviors were influenced by factors such as years of clinical experience, geographical area, and practice setting.

The results point to persistent gaps in the understanding of pharmacological indications, differences in dosing between fever and pain, and the perceived efficacy and safety profiles of paracetamol and ibuprofen. These gaps may contribute to inconsistent treatment practices and potential risks, especially in rural areas or among clinicians with less access to updated clinical training. At the same time, the survey revealed a high level of interest among PCPs in further education—particularly in areas such as dosage guidelines, safety considerations, and the management of adverse effects.

Taken together, these findings underscore the need for an enhanced dissemination of clinical guidelines and for a structured, accessible continuing of medical education programs. A national consensus document and updated, practical guidance could help to unify prescribing practices and ensure a more consistent, evidence-based management of fever and pain in pediatric outpatients. Future research should aim to validate these findings through larger-scale or longitudinal studies, include hospital-based pediatricians, and explore the impact of contextual and systemic factors on clinical decision-making.

## Figures and Tables

**Figure 1 life-15-01048-f001:**
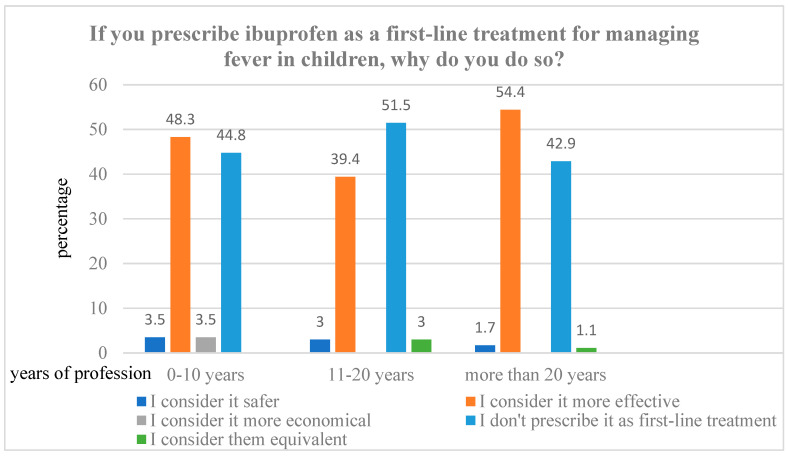
Reason to prescribe ibuprofen as first-line treatment for fever.

**Figure 2 life-15-01048-f002:**
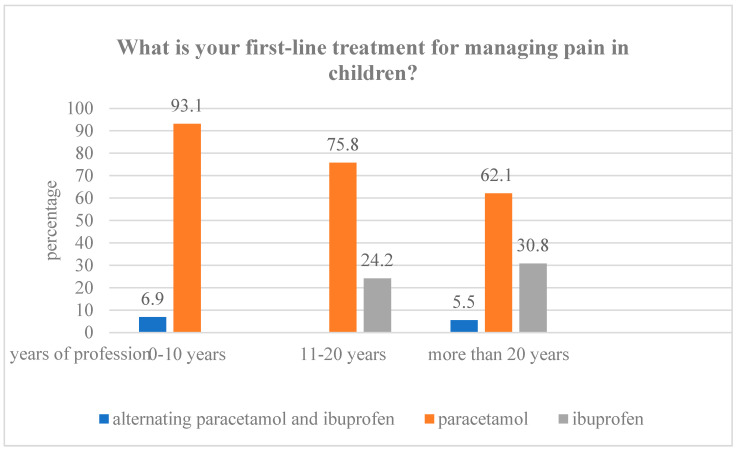
First-line treatment for pain management.

**Table 1 life-15-01048-t001:** Key points of fever management in pediatric patients.

Fever Management	Recommendations
Assessment of Fever	Axillary temperature measurement using a digital thermometer is recommended in children younger than 4 weeks of age in all settings. In the hospital or ambulatory care setting, axillary temperature measurement using a digital thermometer or an infrared thermometer (tympanic or with or without skin contact) is recommended in children older than 4 weeks.
Antipyretics (Acetaminophen/Ibuprofen)	Paracetamol and ibuprofen are the only antipyretic drugs recommended for use in children. Combined or alternating use of ibuprofen and paracetamol is not recommended.
Dosing	Dosing should be based on weight and age guidelines. A total of 10–15 mg/kg of paracetamol.

Adapted from reference [4].

**Table 2 life-15-01048-t002:** WHO pain ladder for pediatric patients.

Pain Level	Treatment Approach
Mild Pain	Non-opioid analgesics (e.g., paracetamol, NSAIDs like ibuprofen)
Moderate Pain	Weak opioids (e.g., codeine, tramadol), with or without paracetamol or an NSAID
Severe Pain	Strong opioids (e.g., morphine), possibly combined with adjunct therapies (e.g., systemic steroids, neuroleptics, anticonvulsants) and anesthetic drugs (e.g., local lidocaine, ketamine)

Adapted from reference [13].

**Table 3 life-15-01048-t003:** Recommended doses of paracetamol and ibuprofen in children. Adapted from references [7,12,14].

Fever		Pain	
Paracetamol	oral administration: 0–3 months: 10 mg/kg every 4–6 h 3 months: 15 mg/kg every 4–6 h rectal administration: 20 mg/kg every 4–6 h	Paracetamol	oral administration: loading dose: 20 mg/kg, then 10–15 mg/kg every 4–6 h rectal administration: 20 mg/kg every 4–6 h
Ibuprofen	oral administration: 3–6 months: 5 mg/kg every 6–8 h >6 months: 7–10 mg/kg every 6–8 h	Ibuprofen	oral administration: 7–10 mg/kg every 6–8 h

**Table 4 life-15-01048-t004:** Paracetamol dosage prescribed by the respondents.

Paracetamol Dosage	Total Number of Respondents	0–10 Years of Clinical Practice	11–20 Years of Clinical Practice	>20 Years of Practice	*p* Value
10–15 mg/kg every 4–6 h	156 (63.9%)	18 (62.1%)	20 (60.6%)	118 (64.8%)	0.76
15–20 mg/kg every 4–6 h	84 (34.4%)	10 (34.5%)	13 (39.4%)	61 (33.5%)	0.76
Unclear answer	4 (1.6%)	1 (3.5%)	0 (0.0%)	3 (1.7%)	

**Table 5 life-15-01048-t005:** Ibuprofen dosage prescribed by respondents.

Ibuprofen Dosage	Total Number of Respondents	0–10 Years of Clinical Practice	11–20 Years of Clinical Practice	>20 Years of Clinical Practice	*p* Value
10–15 mg/kg every 6–8 h	86 (35.2%)	12 (41.4%)	10 (30.3%)	64 (35.2%)	0.83
5–10 mg/kg every 6–8 h	156 (63.9%)	17 (58.6%)	23 (69.7%)	116 (63.7%)	0.83
Unclear answer	2 (0.8%)	0 (0.0%)	0 (0.0%)	2 (1.1%)	

**Table 6 life-15-01048-t006:** Paracetamol and ibuprofen reported adverse effects.

	Total Number of Respondents	0–10 years of Clinical Practice	11–20 Years of Clinical Practice	>20 Years of Clinical Practice	*p*-Value
How often do you encounter adverse effects related to paracetamol in your patients?					<0.05
Never	99 (40.6)	20 (69)	14 (42.4)	65 (35.7)	
Rarely	138 (56.6)	9 (31)	18 (54.6)	111 (61)	
Occasionally	7 (2.9)	0 (0.0)	1 (3)	6 (3.3)	
What are the side effects of paracetamol use in pediatrics that you most often observe in your patients?					0.47
Allergic reactions	86 (35.2)	6 (20.7)	13 (39.4)	67 (36.8)	
None	58 (23.8)	12 (41.4)	7 (21.2)	39 (21.4)	
Gastrointestinal issues	56 (23)	5 (17.2)	7 (21.2)	44 (24.2)	
Liver problems	34 (13.9)	4 (13.8)	4 (12.1)	26 (14.3)	
Other	8 (3.3)	1 (3.5)	2 (6.1)	5 (3.3)	
How often do you encounter adverse effects related to ibuprofen in your patients?					0.28
Rarely	153 (62.7)	22 (75.9)	20 (60.6)	111 (61)	
Occasionally	68 (27.9)	3 (10.3)	11 (33.3)	54 (29.7)	
Never	22 (9)	4 (13.8)	2 (6.1)	16 (8.8)	
Frequently	1 (0.4)	0 (0.0)	0 (0.0)	1 (0.6)	
What are the side effects of ibuprofen use in pediatrics that you most often observe in your patients?					0.62
Gastrointestinal issues	143 (58.6)	18 (62.1)	24 (72.7)	101 (55.5)	
Allergic reactions	79 (32.4)	7 (24.1)	9 (27.3)	63 (34.6)	
Other	5 (2)	2 (6.9)	0 (0.0)	3 (1.7)	
None	10 (4.1)	2 (6.9)	0 (0.0)	8 (4.4)	
Liver problems	7 (2.9)	0 (0.0)	0 (0.0)	7 (3.9)	

## Data Availability

The original contributions presented in this study are included in the article. Further inquiries can be directed to the corresponding author(s).

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
