# Peer review of "Fever and Pain in Italian Children: What Pediatricians Really Do"

_life, 2025, doi:10.3390/life15071048_

Round 1

Reviewer 1 Report

Comments and Suggestions for Authors

METHODS

  • Are there any inclusion and exclusion criteria for the subjects (pediatrician)?
  • “Based on the sample size, the survey had a margin error of 2.55%”: was this based on 900 or 200 subjects?
  • How did the authors select the subjects?

RESULTS

  • Was the proportion of urban area and age valid as representation of Italian situation?
  • What was the highest dosage of paracetamol and ibuprofen recorded from the subjects?
  •  

DISCUSSION

  • Why the response rate was low (20%)?
  • Was it true that the majority of Italian pediatrician more than 55 years of age?
  • There were 3 references in the list (number 9, 12, and 20) which also had similar topic with this study. How was the comparison of the results?
  • Was it true that this study was the first?

TABLES

  • Table 6: The total percentage of never/really/occasionally for total number of respondents was more than 100%
  • Table 6: The total percentage of never/really/occasionally/frequently for >20 years of clinical practice was more than 100%
  • Table 6: The total percentage of gastrointestinal/allergic/other/none/liver for >20 years of clinical practice was more than 100%

FIGURES

  • Figure 1: There was a typo in the title

Author Response

Dear Reviewer 1,

thank you very much for your comments on our manuscript. We have modified our manuscript following your suggestions, in particular:

  • The inclusion criterion for our survey’s partecipants is that partecipants had to be PCPs in current activity on the Italian territory; we have added this sentence in the Methods section.
  • The margin of error was based on the number of respondents (244).
  • The characteristics of our survey population are representative of the National Situation. According to Italian Ministry of Health Database, currently the population of Primary Care Pediatricians is largely composed (77%) of doctors with more than 23 years of seniority of specialization (over 5,100 pediatricians). In Italy, pediatricians are women in 70% of cases.  We have added the specific reference in the text.
  • The highest prescribed dosage of paracetamol was 20 mg/kg, the highest prescribed dosage of ibuprofen was 15 mg/kg, as reported in Appendix A
  • The response rate was low probably because this survey was completed on a voluntary basis and there was no reimbursement nor any facilities related to that.
  • Reference 9 deals with fever management among Pediatricians, reference 12 describes Pediatricians’ attitude towards pain treatment in ER setting, whereas reference 20 describes healthcare professionals’s attitude towards pain and fever management. The results of our survey, when comparable, are in line with those reported in these references.
  • Our study was the first to analyze PCPs’ attitude toward both fever and pain management in outpatients’ setting in Italy.
  • In Table 6, more than one answer was possible to each question, therefore the sum of the percentage of the answers is more than 100%
  • We have corrected Figure 1 title.

Reviewer 2 Report

Comments and Suggestions for Authors

Dear authors, 

I read with interest your article. Great topic selection and a nice try to capture pain and fever management among PCPs. 

It would be nice to include in the discussion section whether there are any workshops/educational seminars on how you handle pediatric pain in Italy. This is a big area of interest as we all know how variable practices are in peds pain management worldwide. 

Also, is there a plan on revising the 2016 Italian guidelines on pain and fever management based on this results? How results of this study will shape future research on the field? Please add a paragraph in the discussion presenting the next steps.

Lastly comparison with other KAP surveys would be beneficial for the discussion as it lacks comparison with available literature.

Best regards

Author Response

Dear Reviewer 2,

thank you very much for your comments on our manuscript. We have modified our manuscript following your suggestions, in particular:

  • We have added a paragraph in the discussion section regarding future reasearch
  • In the discussion section we have specified that there are workshops/educational seminars on how you handle pediatric pain in Italy.

Reviewer 3 Report

Comments and Suggestions for Authors
  • Flaws of style in references are indicated above (comments about reference).
  • Authors are recommended to depict the background of insurance in Italy (e.g. whether prescribing different drugs leads to the economic burden for patients or not) and clarify which is more economical, paracetamol or ibuprofen.

Author Response

Dear Reviewer 3,

Thank you very much for your comments on our manuscript. We have modified our manuscript following your suggestions, in particular:

  • We have modified references as you suggested.
  • We have depicted the background of insurance in Italy.